# The Role of TGF-β Signaling in Saphenous Vein Graft Failure after Peripheral Arterial Disease Bypass Surgery

**DOI:** 10.3390/ijms241210381

**Published:** 2023-06-20

**Authors:** Changhuai He, Pin Ye, Xuecheng Zhang, Elham Esmaeili, Yiqing Li, Ping Lü, Chuanqi Cai

**Affiliations:** Department of Vascular Surgery, Union Hospital, Tongji Medical College, Huazhong University of Science and Technology, Wuhan 430022, China; h15719349225@163.com (C.H.); pinye@hust.edu.cn (P.Y.); zxc420321@hotmail.com (X.Z.); i201720133@hust.edu.cn (E.E.); yiqingli@hust.edu.cn (Y.L.); lvping2013@aliyun.com (P.L.)

**Keywords:** peripheral arterial disease, TGF-β signaling, saphenous vein graft, neointimal, graft failure

## Abstract

Saphenous vein bypass grafting is an effective technique used to treat peripheral arterial disease (PAD). However, restenosis is the major clinical challenge for the graft vessel among people with PAD postoperation. We hypothesize that there is a common culprit behind arterial occlusion and graft restenosis. To investigate this hypothesis, we found TGF-β, a gene specifically upregulated in PAD arteries, by bioinformatics analysis. TGF-β has a wide range of biological activities and plays an important role in vascular remodeling. We discuss the molecular pathway of TGF-β and elucidate its mechanism in vascular remodeling and intimal hyperplasia, including EMT, extracellular matrix deposition, and fibrosis, which are the important pathways contributing to stenosis. Additionally, we present a case report of a patient with graft restenosis linked to the TGF-β pathway. Finally, we discuss the potential applications of targeting the TGF-β pathway in the clinic to improve the long-term patency of vein grafts.

## 1. Introduction

Peripheral arterial disease (PAD) is a condition in which fatty deposits (called plaque) slowly narrow the arteries and affect blood circulation, especially in the legs and feet [1]. More than 200 million people worldwide suffer from PAD; especially with the aging of society, PAD is becoming a serious public health problem [2,3]. The symptoms of PAD depend on the size of the affected blood vessels and the degree of ischemia in the target organs. PAD is progressive, with three different clinical stages: asymptomatic in the early stage, intermittent claudication in the middle stage, and static pain and lower extremity ulcers in the late stage [4]. To salvage the extremities and improve the quality of life among people with PAD, autologous saphenous vein graft (SVG) bypass grafting is one of the common methods used for revascularization [5]. However, the late patency rate of SVG bypass was only between 55% and 60% at the 10-year follow-up [6]. The exact mechanisms of SVG bypass failure remain unclear. Peripheral artery disease bypass grafts (PABGs) are less likely to present atherosclerosis lesions than coronary artery bypass grafts (CABGs), whereas PABGs present more severe intimal hyperplasia [7]. It has been well demonstrated that TGF-β signaling plays an essential role in neointimal hyperplasia which is linked to cellular motility, survival, and proliferation [8]. Increased TGF-β signaling is linked to poor outcomes in the CABGs [9,10]. Margreet [11] reported the distinct atherosclerosis features among the different peripheral arteries, and the progression in the femoral artery was the slowest. Some specific molecular pathways in the femoral artery may accelerate SVG bypass failure. One of the key pathophysiological patterns between PAD and PABG is fibrosis, which is usually linked to TGF-β signaling [12]. In this review, we explored the possible specific signaling pathways for saphenous vein graft failure after bypass surgery in peripheral arterial disease through a combination of bioinformatics, a literature review, and our case report. Additionally, the potential therapeutic strategies were summarized to increase the SVG bypass patency rates.

## 2. Identification and Analysis of Special Pathways

Atherosclerosis is a systemic disease. Although the risk factors and pathogenesis are similar, the pathological manifestations vary from site to site [13]. Whether differentially expressed genes promote a decrease in graft patency remains unknown. With RNA-sequencing data becoming increasingly publicly available, bioinformatics is increasing in importance in clinical research. Through transcriptomic analysis, the link between atherosclerosis and graft patency must be studied. Although the molecular mechanisms underlying the presence of femoral artery specificity remain unclear, we sought to identify differentially expressed genes (DEGs) and potential molecular pathways through bioinformatics. The GSE100927 dataset was obtained from the public GEO database of the American National Center for Biotechnology Information (www.ncbi.nlm.nih.gov/geo) on 10 June 2022. The microarray data of GSE100927 platform, based on the GPL17077 platform (Agilent-039494 Sure Print G3 Human GE v2 8x60K Microarray 039381), and series matrix file(s) were downloaded from the GEO and saved as TXT files. R software (version 4.2.1) was used to process the downloaded files. We selected 23 of these normal artery samples (containing 11 infrapopliteal and 12 femoral arteries) and 40 atherosclerotic samples (containing 14 infrapopliteal and 26 femoral arteries). Heat maps were plotted using the Limma software package (3.56.2) to display all up- and downregulated DEGs in the infrapopliteal and femoral arteries in the normal group and PAD patients, respectively. Adjusted *p*-values of <0.05 and fold change (logFC) of >1 were considered to indicate DEGs. A total of 5 DEGs were identified between the normal infrapopliteal arteries and normal femoral arteries (Figure 1A) and 37 were identified between the atherosclerotic infrapopliteal arteries and atherosclerotic femoral arteries (Figure 1B). *Dusp1* and *Serpine1* were upregulated in both the normal and atherosclerotic femoral samples compared with the infrapopliteal arteries. The functions of these two key hub genes and the KEGG pathway are shown in Table 1. The STRING database (https://cn.string-db.org/) is a precomputed worldwide resource for the exploration and analysis of interactions between known and predicted protein–protein interactions. The related genes, DUSP1 and SERPINE1, were separately selected on 25 June 2022, and the correlations between DUSP1 and SERPINE1 hub genes were visualized using the free visualization software tool Cytoscape 3.9.1 (Figure 1C). Both DUSP1 and SERPINE1 play important roles in the TGF-β signaling pathway, as shown by protein interaction networks and the KEGG analysis of genes.

We determined that the TGF-β pathway plays a substantial role in different healthy lower extremity arteries. Then, we wanted to determine whether the TGF-β pathway also has an important role in peripheral atherosclerotic disease. We further analyzed the expression of TGF-β between people with PAD and healthy people groups. Liao reported 54 genes associated with TGF-β from MSigDB (v7.4) [14]. To determine the expressions of TGF-β-related genes between the PAD and control groups, a gene box map was created using R software (4.1.2). Among the reported 54 TGF-β-related genes, 40 TGF-β-related genes were found to be meaningfully expressed in both groups (Figure 2). The TGF-β signaling pathway is crucial in regulating immune homeostasis, lipid metabolism, and inflammatory responses in the vasculature [15,16]. By studying the role of TGF-β, we aimed to increase the patency rate of vein transplantation. Additionally, we summarized the studies on TGF-β in vascular remodeling through a literature review.

## 3. TGF-β Family and Signaling Pathway

The TGF-β cytokine family is ubiquitous and plays an important role in inflammation, repair, and immunity; this family is essential for the development of the body. In general, TGF-β has a stimulatory effect on the cells of mesenchymal origin and an inhibitory effect on the cells of epithelial/neuroectodermal origin. Dysregulation of the TGF-β downstream pathway leads to the development and progression of cancer and fibrotic diseases [17,18]. TGF-β belongs to the peptide family and has three types of receptors (β1: homodimer; β2: homodimer; β3: heterodimer), which play different roles according to the effects of different ligand signals. Among them, TGF-β1 has been extensively studied due to its important role in immunity and fibrosis [19]. TGF-β2 and TGF-β3 are less expressed in mammals, but TGF-β2 plays an important role in glucose and fatty acid metabolism. Because TGF-β3 is similar to TGF-β1 in immunity, it is expected to be a focus in future disease research [20,21]. The two main TGF-β signaling pathways are the canonical (Smad-dependent) and noncanonical (Smad-independent) pathways. The canonical (Smad-dependent) signaling pathway is highly conserved. The ligand binds to the transmembrane receptor on the cell surface to form an active TGF-β type II receptor and phosphorylates the GS domain (rich in glycine and serine residues) in TGF-β1. After GS phosphorylation, it recruits the receptors regulating Smad2 and Samd3. R-Smads form an active complex with Smad4 and transport it to the nucleus to regulate the transcription of target genes. Smad7 may be recruited into the complex of activated Smad2/3 to initiate their degradation by Smad-specific ligases, which, in turn, prevents them from being transcribed. Among the noncanonical pathways, the diversity of intracellular TGF-β signaling depends not only on the type of ligand, receptor, Smad mediator or Smad-interacting protein but also on the ability of TGF-β to activate other signaling pathways, such as P38, MAPKs, ERK, AKT, and so on [22,23,24]. Both the canonical and noncanonical pathways have been extensively discussed and validated (Figure 3).

## 4. TGF-β Signaling Mediates Endothelial–Mesenchymal Transition (EMT) during Vein Graft Remodeling

There were several studies that liked the saphenous vein under the physiological shear stress circumstances (Table 2). The varicose vein was linked to dysregulation of TGF-β1 signaling [25,26,27,28,29]. However, the target vein grafts in the PAD bypass surgery would avoid the varicose veins. From cell culture studies to tissue studies, TGF-β signaling played an essential role in vascular remodeling [30,31,32,33].

Under vascular injury circumstances, vascular remodeling and intimal hyperplasia are normal physiological responses to adapt to the new organismal environment and are usually self-limiting processes. However, in the majority of venous grafts, abnormal venous stenosis and excessive vascular remodeling are observed. The pathophysiological vascular remodel is linked to endothelial–mesenchymal transition (EMT), and EMT is a key feature of venous graft remodeling [34,35]. The mechanism of EMT in the process of neointima formation and intimal hyperplasia is unclear. TGF-β plays a central role in fibrosis by regulating the phenotype and function of fibroblasts. Endothelial cells undergoing EMT acquire the phenotype of a fibroproliferative stroma and exhibit abnormal migration and proliferation [36]. Endothelial cells (ECs) can give rise to myofibroblasts, acquire a mesenchymal phenotype, and express α-SMA, waveform proteins, and collagen. Additionally, EMT may be an important source of mesenchymal cells, with TGF-β further contributing to fibrosis progression by inducing EMT. Byambasuren revealed that miR-374b levels are elevated in human coronary artery disease and inversely correlate with MAPK7 expression, suggesting that the TGF–miR-374b–MAPK7 axis plays a key role in the induction of EMT during intimal hyperplasia and early lesions [37]. By comparing the induction of TGF-β for EMT with the expression of miR-92a, the neointimal formation was found to be not entirely dependent on the phenotypic conversion of vascular smooth muscle cells; it was also associated with EMT [38]. EMT and thrombosis may be mutually reinforcing, and thrombin may exacerbate EMT through TGF-β/Smad3 [39]. In large animal vascular injury experiments, veins exhibited intimal thickening, associated with the features of EMT, including increased TGF-β and Smad expression, loss of endothelial and gain of mesenchymal marker expression, and coexpression of endothelial and mesenchymal markers intimal cells, suggesting that endothelium has strong plasticity in the process of membrane proliferation [40]. The mechanisms of the TGF-β induction of EMT and the regulation of pathophysiological changes are unknown at this stage, but targeting EMT may be a potential strategy for the prevention and treatment of endothelial hyperplasia in vascular remodeling diseases.

## 5. Extracellular Matrix

TGF-β is an important regulator of the extracellular matrix, performing multiple functions depending on the signals encountered. TGF-β stimulates the production of proteoglycans, collagen, and fibronectin in the extracellular matrix, and regulates the secretion of protease inhibitors, leading to excessive ECM deposition and graft stenosis [41,42]. Plasmin and matrix metalloproteinases (MMPs) play important roles in intimal proliferation, and they interact with TGF-β to regulate extracellular matrix remodeling [43]. The activation of fibrin by u-PA and t-PA not only degrades fibrin but also activates TGF-β and MMP. Activated TGF-β then induces the plasminogen activator inhibitor (PAI), which forms negative feedback with the activation of fibrin and maintains the stability of the extracellular matrix [44,45]. Changes in u-PA and t-PA levels have important implications for restenosis after venous grafting. u-PA expression is enhanced in rats in which graft restenosis occurs compared with normal rats, and intimal hyperplasia is significantly increased in grafted vessels when u-PA is placed around the damaged vessel in the form of a pluronic gel [46]. The activation of TGF-β, which is dependent on increased MMP2 activity, may also increase the secretion of protease inhibitory factor (TIMP-1) [47]. TGF-β is highly expressed in the neointima and increases the deposition of extracellular matrix and the progression of fibrosis by regulating the expression of MMPs and TIMPs, inducing collagen production [43]. In addition, TGF-β1 inhibits rt-PA-mediated induction of MMP-2 and MMP-9, and inhibiting rt-PA may be another mechanism by which the protective effect of TGF-β1 occurs [48]. The activity of MMP-2 in normal blood vessels is stable, whereas the expression of MMP-2 increases after vascular injury as well as senescence. By reducing vascular TGF-β levels and MMP-2 activity in rats, vascular remodeling was thus improved and collagen deposition was reduced [49]. Collagen occupies an important place in the extracellular matrix, and large numbers of type 1 and type 3 collagen fibers are seen in advanced hyperplastic endothelium [50]. Sun et al. revealed the dynamic expression of MMP-1 and TIMP-1 through TGF-β in the grafted veins of rats and demonstrated that TGF-β1 plays an indispensable role in the deposition of collagen fibers after vascular injury [51]. Angiotensin II (Ang II) is responsible for the pathophysiology of vascular fibrosis and can affect the synthesis and activation of TGF-β, MMP, and their tissue inhibitors [52]. TGF-β interacts with related protease pathways, which play an important role in cell morphogenesis, proliferation, and differentiation and are needed for the precise regulation of endothelial proliferation.

## 6. Fibrosis

TGF-β induces and regulates vascular remodeling and wound healing upon injury, but dysregulated signaling contributes to the development of fibrosis [53]. Fibroblast growth factor (FGF) is a peptide growth factor of the heparin-binding growth family that can be synthesized by endothelial and vascular smooth muscle cells [54]. Acidic fibroblast growth factor (aFGF) and basic fibroblast growth factor (bFGF) share the same receptor, but bFGF is 30–100 times more potent than aFGF and is one of the growth factors that directly act on vascular cells to induce endothelial cell growth and angiogenesis [55]. FGF enhances VSMC proliferation and intimal hyperplasia in genetically deficient mice [56]. Inserting the FGF gene into the arterial wall revealed numerous neointimal angiogenesis and promoted intimal hyperplasia [57]. In laboratory rat balloon-stripped carotid arteries, bFGF’s effect on vessels was linearly related to the amount of SMC proliferation within the blood vessel and could be similarly modulated by the products of local injury and/or factors in the vessel wall [58]. When FGF synthesis was inhibited by shRNA knockdown, intimal hyperplasia could be effectively alleviated [59]. FGF is a potent mitogen for vascular smooth muscle and endothelial cells and plays an important role in vein graft failure. In in vitro experiments, IL-6 and TGF-β induced the expression of FGF receptors. The expression of aFGF and its receptor were observed in cardiac allograft vessels [60]. TGF-β effectively inhibits FGF stimulation in endothelial cells to reduce intimal hyperplasia, and TGF-β induces endothelial cells to form myofibroblasts and other types of mesenchymal (non-myofibroblastic) cells [61]. After vascular injury, FGFs and cytokines bind to their receptors and then trigger the activation of signaling pathways and transcription factors in a Smad-dependent or Smad-independent manner. The dysregulation of the TGF-β1/Smad pathway is an important pathogenic mechanism of tissue fibrosis. Smad2 and Smad3 are the two main downstream regulators that promote TGF-β1-mediated tissue fibrosis, whereas Smad7 acts as a negative feedback regulator of the TGF-β1/Smad pathway, thereby protecting against TGF-β1-mediated fibrosis [62]. TGF-β induces the transition from fibroblasts to myofibroblasts and participates in the regulation of fibrotic gene expression together with related transcription factors to aggravate the degree of fibrosis [63]. In experimental studies of vascular injury in rats, TGF-β played an important role in mediating vascular fibrosis and vascular remodeling after injury [64]. Activating transcription factor 3 (ATF3) plays a key role in fibroblast activation and induces activation of the TGF-β/Smad signaling pathway in fibroblasts [65]. TGF-β can also regulate the extent of fibrosis by recruiting and activating monocytes and fibroblasts and by inducing gene expression in the ECM. The specific role of targeting TGF-β in the pathogenesis of fibrosis has been demonstrated in animal and clinical experiments in other diseases; the mechanism of fibrosis development in grafts remains to be experimentally demonstrated [66,67].

## 7. TGF-β Signaling and Intimal Hyperplasia

Despite rapid advances in endovascular therapy, autologous vein bypass remains the most effective and durable revascularization strategy for people with PAD. The preservation of vein graft patency has always been a hot research topic, but the rate of early and long-term graft patency has remained unchanged over the past few decades [68]. Three main processes, thrombosis, intimal hyperplasia, and atherosclerosis, lead to vein graft failure. Of these, intimal hyperplasia is critical for the long-term patency of venous grafts [69]. Carrel first described the pathological manifestations of intimal hyperplasia after vein graft surgery. Intimal hyperplasia after a vascular injury has long been a major problem for researchers. Under physiological conditions, intimal hyperplasia marks the healing of wounds; however, why do some vessels persistently develop stenosis and occlusion in the postinjury period? As shown in Figure 4, pathological intimal hyperplasia begins immediately in damaged blood vessels and continues to progress. Vascular endothelium injury rapidly causes platelet adhesion and aggregation and, with the activation of thrombin and tissue factor, intimal hyperplasia and thrombosis occur [70]. Leukocytes and adhesion molecules adhere to damaged endothelial cells under the action of chemokines and stimulate endothelial cells to produce various cytokines and growth factors, which interact with proteases, having a notable regulatory effect on the extracellular matrix and vascular smooth muscle cells [71]. Abnormal migration and proliferation of vascular smooth muscle cells, reduction in endothelial progenitor cells, and thrombosis, accompanied by extracellular matrix deposition and inflammatory infiltration, are pathological changes that lead to constant lumen narrowing and venous graft failure [71]. The results of animal studies have shown that TGF-β is upregulated at the site of vascular injury (Table 3). Through targeted inhibition of TGF-β, the investigators found that intimal hyperplasia was effectively inhibited in animal models. Brilakis reviewed the natural history of the SVGs, emphasizing the importance of the native saphenous vein for improving graft patency [72]. With the recent exploration of intimal hyperplasia at the molecular level, TGF-β was found to have not only biological functions in inflammation, tissue repair, and embryonic development, but also crucial regulatory roles in cell growth, differentiation, and immune function, indicating a potential therapeutic strategy to inhibit intimal hyperplasia [17].

## 8. Risk Factors of Bypass Graft Failure

In clinical practice, patients with combined systemic vascular diseases such as diabetes and hyperlipidemia have a higher susceptibility to lower limb ischemia, which is possibly related to the molecular mechanisms of accelerated vascular fibrosis and intimal hyperplasia [85,86,87,88]. Diabetes can cause a high rate of vascular damage, with intimal hyperplasia, stenosis, and subsequent fibrinoid degeneration of the vessels from small arteries. A total of 20% of people with PAD have a combination of diabetes and are more likely to develop intimal hyperplasia and restenosis of the vessels after surgery than healthy people [85,89,90]. Here, the main mechanism is insulin resistance, leading to endothelial insufficiency and accelerated platelet aggregation, which activates growth factors and promotes the proliferation of smooth muscle cells and inflammatory cells, thus causing intimal hyperplasia [91]. TGF-b can antagonize BMP-6 signaling and regulate smooth muscle cell differentiation, leading to enhanced intimal hyperplasia, as found in a mouse model of diabetes, by regulating circulating smooth muscle progenitor cells (SPCs), and TGF-β/BMP-6 expression promotes intimal hyperplasia in mice [92]. Connective tissue growth factor (CTGF) is a potent profibrotic factor and some of the downstream mediators of TGF-β. Modulating the TGF-β signaling pathway by targeting CTGF interference can help prevent intimal hyperplasia in diabetic large vessels [93]. An analysis of clinical data suggests that TGF-β1 may be a risk factor for endothelial hyperplasia and that restenosis is more severe in people with concomitant diabetes [94]. Advanced glycation end products (AGEs) accumulate with age in vascular tissues. AGEs increase extracellular matrix production by upregulating TGF-β and can interact with RAGs, which may lead to smooth muscle cell proliferation, extracellular matrix degradation, and the eventual production of intimal hyperplasia and restenosis [95]. In addition, genetic, surgical operation, inflammation, and hemodynamic factors are important risk factors.

## 9. Case Presentation

A 50-year-old woman developed intermittent claudication in her right extremity without obvious cause prior to presentation at our hospital, denying abdominal pain, nausea, and vomiting. The physical examination showed pulselessness of the right dorsal artery. The skin temperature of the right foot was lower than that of the other side. She complained of pain, pallor, and paralysis after walking longer than 50 m in her right foot. Her blood pressure was 130/70 mmHg. Laboratory evaluation revealed an LDH level of 1060 U/L, mild leukocytosis (11.29 × 10^9^/L) with 88.3% neutrophils, slight renal dysfunction (creatinine 121 µmol/L and urea nitrogen 7.0 mmol/L), and a CRP level of 134 mg/L. Routine testing revealed that urine, coagulation function, autoantibodies, and other indicators were all normal. The patient denied a past medical history of vascular intervention or recent trauma. However, she had a history of smoking. Consequently, a computerized tomography angiogram (CTA) revealed right femoral artery occlusion, and the runoff artery was blocked (Figure 5A). After adequate discussion with the patient and her family, great saphenous vein bypass surgery was performed (Figure 5B). Five days later, the patient was discharged, which was accompanied by notable prolonged walking distance ability, and the pallor, pain, and paralysis of her right foot were substantially alleviated. During the one-year follow-up, the patient complained of serious pain in the surgical extremity. The physical examination revealed pulselessness in the graft vein and thrombophlebitis. Fortunately, we found no obvious ischemia in the leg. We performed a thrombectomy for the graft vein. During the surgery, we found a lack of blood flow in the graft vein. After acquiring her informed consent, we collected one segment of the graft vein. The postoperative histopathological analysis showed obvious inward remodeling in the graft vein accompanied by extensive neointimal hyperplasia and a tiny vessel lumen (Figure 5C). The VVG staining proved that the elastic fiber was broken (Figure 5D), and the positive blue staining in the Masson trichrome staining of the vessel wall was extensive (Figure 5E).

Moreover, we noted α-SMA-positive cells in the whole vessel wall (Figure 6A). The positive TGF-β1 staining (Figure 6B) was observed almost everywhere, in agreement with Masson’s trichrome staining. This implied that the TGF-β1 signaling may be linked to several cellular types. Further experiments need to be designed to reveal the complex cellular biology.

## 10. Clinical Treatment Strategies

All graft veins undergo a series of dynamic structural changes that we call vascular remodeling. Unfortunately, no specific treatment exists for GSV graft stenosis. By linking histological and molecular changes in veins, awareness has been growing of the clinical significance of TGF-β for vascular remodeling. TGF-β inhibitors are now widely used with success in the cancer field [96,97]. Some results have also been obtained in extensive basic trials in the prevention of intimal hyperplasia, so TGF-β appears to be promising in intimal hyperplastic disease [77,98,99]. We extracted data from ClinicalTrials.gov (accessed on 15 June 2023) regarding vascular injury and TGF-β, which are compiled in Table 4. In clinical practice, both pirfenidone and nintedanib slow the progression of disease fibrosis, and their main effect is on the Smad TGF-β pathway. TGF-β facilitates the accumulation of extracellular matrix. Rapamycin and mycophenolate mofetil, novel immunosuppressive agents, reduce profibrotic gene expression in both experimental and clinical settings and are currently showing promise for application [41,100]. Chymase is an important enzyme for the production of Ang II and TGF-β. Chymase inhibitor treatment leads to reductions in Ang II and TGF-β1 expressions, resulting in a marked inhibition of neointimal formation. Chymase inhibition may be a new strategy that can be used to prevent intimal hyperplasia in the clinical setting [101]. Resveratrol is a promising natural antistenosis agent that blocks the TGFβ/Smad3 pathway and can be topically applied intraoperatively to reduce intimal hyperplasia [76]. As TGF-β has a wide range of actions and can mediate many pathways, treatment of intimal hyperplasia with TGF-β needs to be focused on specific pathways to avoid side effects, so more clinical trials are needed to validate its efficacy.

## 11. Prospects

Programmed cell death is a genetically determined, active, and ordered form of cell death that plays an important role in maintaining tissue homeostasis. Many vascular malformations share aberrant molecular signaling pathways with cancers and inflammatory disorders [102]. The induction of cell death has been extensively studied in the field of cancer with good results. However, in recent years, interactions between different cell death mechanisms and neovascularization have been revealed [103]. TGF-β is involved in the control of cell growth, apoptosis, differentiation, and proliferation and plays a central role in endothelial cell signaling and angiogenesis. The role of ferroptosis and pyroptotic death in relation to TGF-β is currently poorly studied in the cardiovascular field, but the different programmed cell deaths do not operate independently; for example, apoptosis, pyroptosis, and necroptosis constitute a single, coordinated cell death system, where one pathway can flexibly compensate for another [104]. We found that TGF-β-related genes are abnormally expressed in peripheral arterial disease and are important in peripheral vein graft failure; they are also strongly associated with programmed cell death. We think that people with peripheral arterial disease may also show abnormal expression of TGF-β-related genes in their veins, leading to a high rate of vein graft failure, an area where basic research is lacking. Additionally, CABG failure is another difficult clinical problem that leaves clinicians at their wits’ end. There may be a similar mechanism in the CABG stenosis. However, in the future, we encourage research, trials, and experiments in the area of programmed cell death in autochthonous veins and endothelial hyperplasia to provide a new direction for increasing the graft vein patency rate. The TGF-β signaling blockade drugs would benefit the patency of PAD bypass grafts. More attention should be put on graft patency maintenance. Moreover, chronic venous insufficiency patients may also benefit from the TGF-β signaling blockade drugs development.

## 12. Conclusions

The failure rate of saphenous vein graft in people with PAD remains high for many years, which is a challenge faced worldwide. Intimal hyperplasia is the main process leading to graft failure. Researchers are increasingly focusing on the molecular level. TGF-β plays an important role in intimal hyperplasia, mediating multiple processes, and the understanding of the importance of its role is increasing (Figure 7). Therapy targeting TGF-β in the graft vein is expected to reduce the degree of intimal hyperplasia of patients in clinical practice, improve the long-term patency rate of vein grafts, and improve the quality of life of patients.

## Figures and Tables

**Figure 1 ijms-24-10381-f001:**
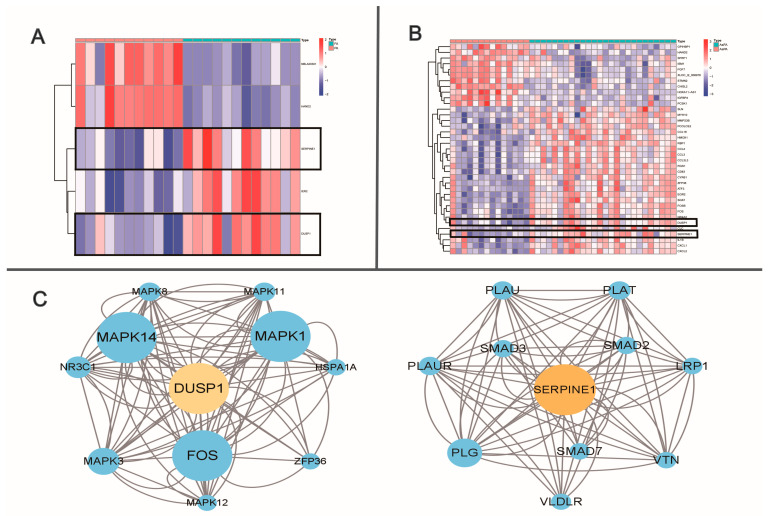
Bioinformatics analysis of infrapopliteal and femoral artery. (**A**) Heat map plot of differentially expressed genes between normal infrapopliteal and normal femoral artery. (**B**) Heat map plot of differentially expressed genes between atherosclerotic infrapopliteal and atherosclerotic femoral artery. Heat map: remarkable DEGs according to adjusted *p*-value and logFC. Red indicates higher gene expression, and green indicates lower gene expression. (**C**) DEG hub gene cluster and network.

**Figure 2 ijms-24-10381-f002:**
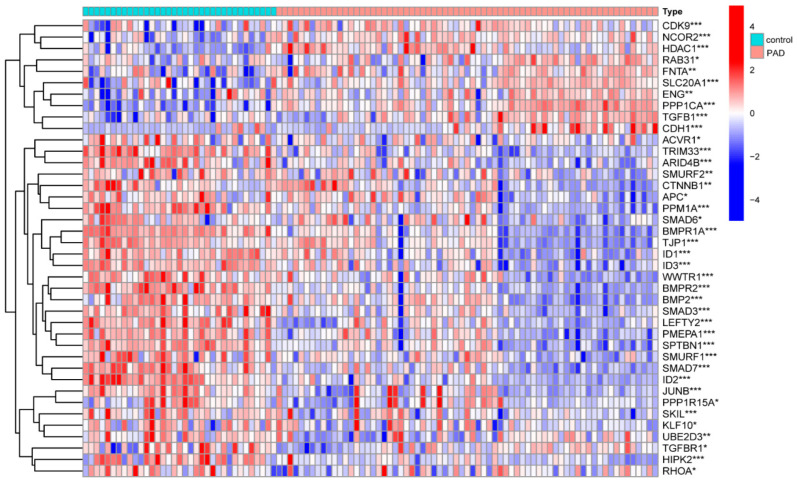
Bioinformatics analysis of TGF-β-related genes in PAD and control vessels. Differential analysis of TGF-β-related genes in PAD and normal vascular tissues. *** *p* < 0.001, ** *p* < 0.01, and * *p* < 0.05.

**Figure 3 ijms-24-10381-f003:**
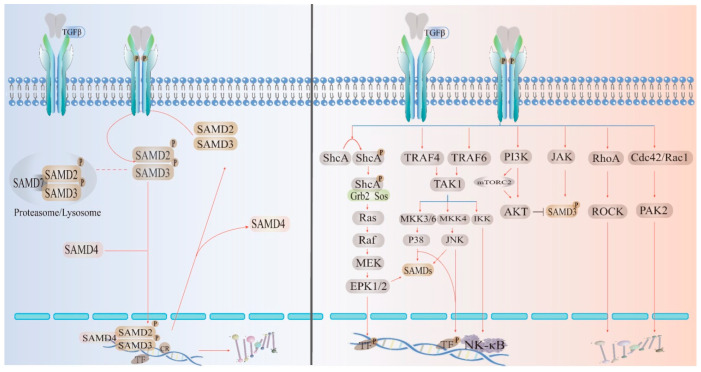
TGFβ signaling pathways: canonical (Smad-dependent) and noncanonical (Smad-independent) pathways. Mature TGF-β dimers bind to two receptors to activate canonical or noncanonical pathways. Ligand activation leads to phosphorylation of pSmad; receptor-activated Smads (R-Smads) interact with Co-Smads and transport them to the nucleus to regulate transcription of target genes. Specific transcription factors lead to cellular responses through effects on target gene expression.

**Figure 4 ijms-24-10381-f004:**
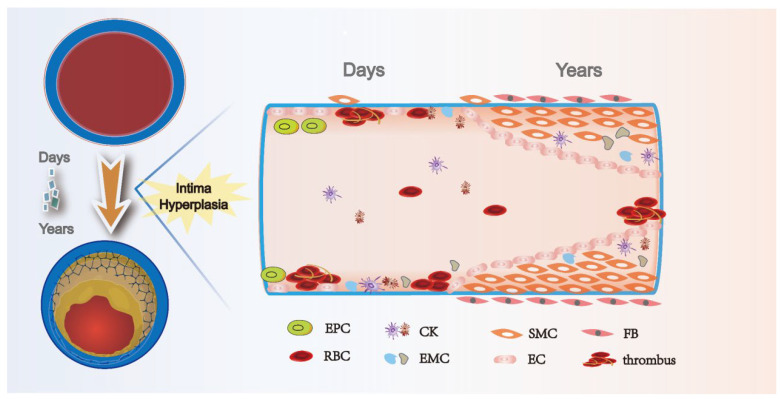
Time course of intimal hyperplasia after vein graft. Vein graft procedure leads to endothelial injury, which immediately results in platelets and thrombosis. Over time, inflammatory cells attach and infiltrate the fibrin layer and intima. Secretion of cytokines, including PDGF and TGF-β, activated smooth muscle cells in the media, and fibroblasts in the adventitia, start migrating toward the intima and induce extracellular matrix deposition, forming the intimal hyperplasia. EPC: endothelial progenitor cell; RBC: red blood cell; CK: cytokines; EMC: extracellular matrix cell; SMC: smooth muscle cell; EC: endothelial cell; and FB: fibroblast.

**Figure 5 ijms-24-10381-f005:**
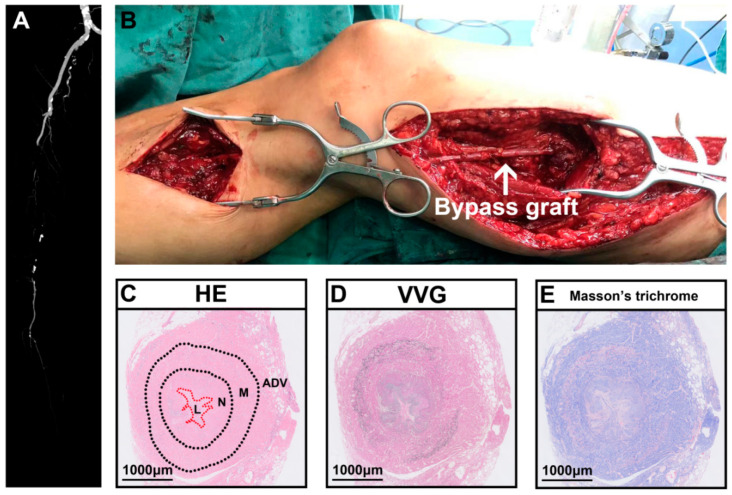
Representative case of saphenous vein graft failure after bypass surgery for peripheral arterial disease. (**A**) Preoperative computerized tomography angiogram (CTA) of person with PAD. (**B**) Intraoperative saphenous vein bypass surgery. (**C**) Representative hematoxylin and eosin (HE) staining showed obvious inward vascular remodeling accompanying narrowed vessel lumen. (**D**) Representative VVG staining showed broken elastic fibers. (**E**) Representative Masson’s trichrome staining showed obvious fibrosis in vessel wall. HE, hematoxylin and eosin; VVG, Verhoeff–Van Gieson staining; ADV, adventitial layer; M, media layer; N, neointimal layer; L, lumen. White arrow indicates bypass graft.

**Figure 6 ijms-24-10381-f006:**
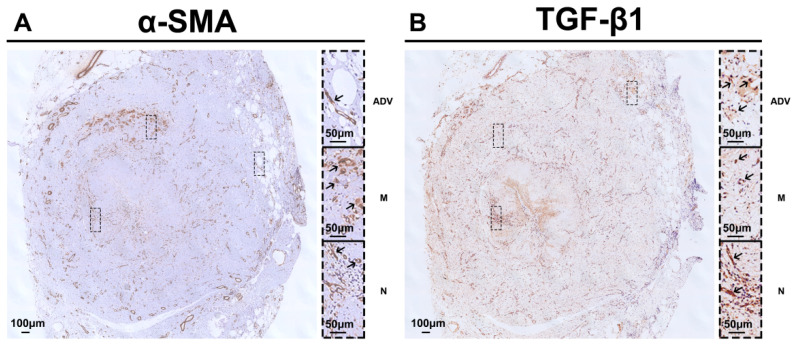
Representative immunohistochemistry staining of saphenous vein graft failure after bypass surgery for peripheral arterial disease. (**A**) Obvious α-SMA(+) cells in the whole vessel wall. (**B**) Abundant positive TGF-β1 staining in the vessel wall from the neointimal layer to the media and adventitial layers. TGF-β1, transforming growth factor β1; ADV, adventitial layer; M, media layer; N, neointimal layer. Black arrows indicate positive staining.

**Figure 7 ijms-24-10381-f007:**
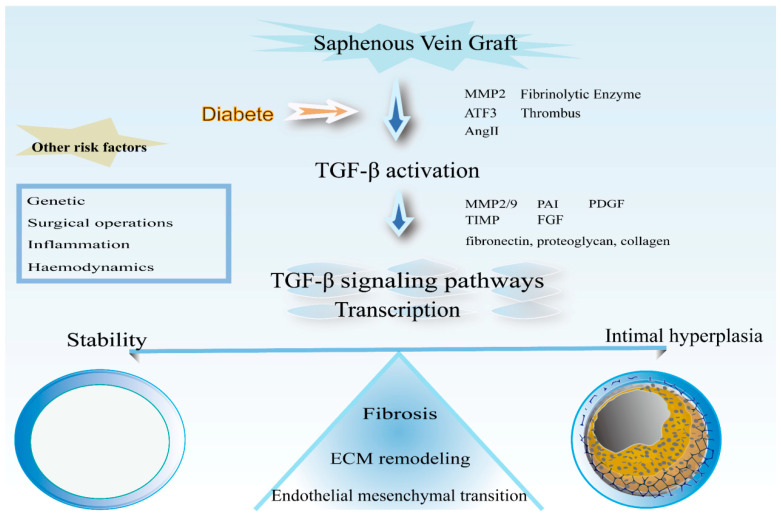
Schematic representation of the processes involved in TGF-β in vascular grafts. Precipitation of extracellular matrix, proliferation, and migration of smooth muscle cells; fibrosis and mesenchymal transformation of endothelial cells are the main processes of intimal hyperplasia after venous graft surgery.

**Table 1 ijms-24-10381-t001:** Functions of two key hub genes and KEGG pathway.

Genes	Function	Super Pathway
*Dusp1*	Dual specificity phosphatase that dephosphorylates MAP kinase MAPK1/ERK2 on both ‘Thr-183’ and ‘Tyr-185’	RAF/MAP kinase cascade
		MAP kinase signaling
		Neuropathic pain signaling in dorsal horn neurons
		ERK signaling
		TGF-β pathway
*Serpine1*	Primary inhibitor of tissue-type plasminogen activator (PLAT) and urokinase-type plasminogen activator (PLAU)	Response to elevated platelet cytosolic Ca^2+^
		Defects of contact activation system (CAS) and kallikrein/kinin system (KKS)
		Gene expression (transcription)
		Circadian clock
		Signaling by TGF-β family members

**Table 2 ijms-24-10381-t002:** TGF-β signaling studies in the great saphenous vein.

No.	Author	Source	Research Content	Conclusion
1	Pedro Serralheiro [30]	Healthy saphenous vein sample	TGF-β1 modulates MMP, TIMP	Involvement of TGF-β1 in the vein wall pathology.
2	Sarah J George [31]	Segments of human saphenous vein	TGF-β1 modulates smooth muscle cell proliferation or apoptosis	Plasmin, generated by uPA, appears to be an important activator of endogenous latent TGF-β.
3	Pedro Serralheiro [32]	Saphenous vein samples	TGF-β1 modulates MMP, TIMP	Decreased expression of TGFβRs might suggest a reduction in TGF-β1 participation in the MMP/TIMP imbalance throughout CVeD progression.
4	Gemma Pascual [25]	Saphenous veins in patients with varicose veins and normal subjects	TGF-β1 modulates chronic venous insufficiency	Aging and varicose pathology induce dysregulation of TGF-β1 that could play an important role in the fibrous process.
5	Ji-Chang Wang [26]	Saphenous veins in patients with varicose veins and normal subjects	PGE2 modulates MMP, TGF-β	PGE2 can affect the remodeling of the extracellular matrix and reduce the elasticity of the vascular walls by promoting the synthesis of TGF-β1 and MMP-1.
6	Julia Buján [27]	Saphenous veins in patients with varicose veins and normal subjects	Varicose veins specimens showed greater LTBP-2 and TGF expression	Varicose veins specimens showed greater LTBP-2 and TGF expression.
7	Theresa Jacob [28]	Saphenous veins in patients with varicose veins and normal subjects	TGF-β1 modulates iNOS	The increased expression of TGF-β1 and presence of macrophages, correlating with overproduction of iNOS, may be associated with varicosity development and deserves further study.
8	Radoslaw Kowalewski [29]	Saphenous veins in patients with varicose veins and normal subjects	TGF-β RII modulates TGF-β	Increased TGF-β RII expression and activation in the wall of varicose veins may be involved in extracellular matrix remodeling related to TGF-β1 and supports its role in the disease pathogenesis.
9	S Mii [33]	SMC derived from human saphenous vein	TGF-β modulates PDGF, b-FGF	TGF-β inhibits both migration and proliferation of human SMC.

**Table 3 ijms-24-10381-t003:** Summary of animal intimal hyperplasia studies to target TGF-β after vascular injury (↓ means decreased, ↑ means increased).

Animal Model	Intervention	Result	Reference
Rats: myocardial infarction	TNAP ↓ → TGF-β1 ↓, α-SMA ↓, Vimentin ↓, and fibronectin ↓	Provides direct evidence that inhibition of TNAP is a novel regulator in cardiac fibrosis and exerts antifibrotic effect mainly through AMPK-TGF-β1/Smads and p53 signals.	Gao et al., 2020 [73]
Rats: carotid artery after balloon injury	Osthole → TGF-β1 ↓, p-Smad2 ↓, TNF-α ↓, IL-1β ↓, NF-κB ↓	Osthole inhibited neointimal hyperplasia in balloon-induced rat carotid artery injury, and the mechanism may involve NF-κB, IL-1β, and TNF-α downregulation, and TGF-β1/Smad2 signaling pathway inhibition.	Li et al., 2017 [74]
Mice and human vasculopathy samples	LMO7 ↓ → TGF-β ↑, SMAD3 ↑	LMO7 induced by TGF-β and limits vascular fibrotic responses through negative feedback regulation of TGF-β pathway.	Xie et al., 2019 [53]
Rats: carotid balloon injury	ERK MAPK ↓ → TGF-β/Smad3 ↓ → VSMC proliferation ↓	TGF-β enhances VSMC proliferation through a Smad3/ERK MAPK signaling pathway.	Suwanabol et al., 2012 [75]
Rats: carotid injury	Resveratrol-(Akt-mTOR pathway) → TGF-β/Smad3 ↓	Resveratrol produces durable inhibition of all three prorestenotic pathologies; a rare feat among existing antirestenotic methods.	Zhu et al., 2017 [76]
Rabbits: balloon-injured carotid arteries	HUK→ TGF-β1/Smad2/3 ↓ → eNOS ↑	HUK attenuates atherosclerosis formation and inhibits intimal hyperplasia by downregulating TGF-β1 expression and Smad2/3 phosphorylation, upregulating eNOS activity.	Lan et al., 2016 [77]
Rats: balloon-injured carotid arteries	Antisense Smad3 → collagen(1, 3)	Transfection of VSMCs with antisense Smad3 can reduce the secretion of type I and III collagen, which then inhibits intimal hyperplasia.	Lu et al., 2012 [78]
Rats: atherosclerosis	Leech → p38MAPK ↓ → TGF-β ↓, blood lipid ↓	Leech may affect the p38MAPK signaling pathway to inhibit proliferation and promote apoptosis of VSMCs via reducing blood lipid levels and suppressing TGF-β.	Wu et al., 2017 [79]
Rats: injured arterial wall	TGF-β/Smad3 ↑ → SMC migration, CXCR4 ↑	Upregulation of TGFβ/Smad3 in injured arteries induces local SMC CXCR4 expression and cell migration, and, consequently, IH.	Shi et al., 2016 [80]
Rats: carotid artery injury	HNK-MSNPs → Smad3 ↓	Honokiol (HNK), a small-molecule polyphenol, can inhibit neointimal hyperplasia after balloon injury.	Wei et al., 2020 [81]
Mice: carotid artery injury	PAI-1 ↓ → TGF-β ↓	PAI-1 is both a critical mediator of TGF-beta1-induced intimal growth and a key negative regulator of TGF-beta1 expression in the artery wall	Otsuka et al., 2006 [82]
Rats: aortals	Mfn2 ↑ → TGF-β ↓	Mfn2 influences TGF-β/Smad pathway and may function as potential chronic rejection inhibitor.	Sun et al., 2019 [83]
Rabbit: femoral and iliac arteries	Furin-like PCs → TGF-β ↑	Furin-like PCs are involved in arterial response to injury possibly through activation of TGF-beta–Smad signaling pathway.	Sluijter et al., 2005 [84]

**Table 4 ijms-24-10381-t004:** Summary of clinical trials about vascular remodeling and TGF-β.

NCT Number	Conditions	Brief Title	Start Date	Phase
NCT02842424	Peripheral arterial disease	Ramipril Treatment of Claudication	July 2020	Phase 4
NCT00382213	Atherosclerosis	A Randomized, Double-blind Study To Compare The Effects Of Olmesartan Medoxomil Versus Placebo In Patients With Established Atherosclerosis	June 2000	Phase 3
NCT02632877	Peripheral arterial disease	Efficacy of Pirfenidone Plus MODD in Diabetic Foot Ulcers	January 2014	Phase 1, Phase 2
NCT04489251	Pulmonary arterial hypertension	Assessment of the TGF-beta Pathway and Micro-RNA in Pediatric Pulmonary Arterial Hypertension	July 2020	-
NCT05269849	Hereditary hemorrhagic telangiectasia	Sirolimus for Nosebleeds in HHT	March 2022	Phase 2
NCT04998227	Acute coronary syndrome	Latent TGF-β-binding Proteins Affect the Fibrotic Process in Renal Impairment and Cardiac Dysfunction	August 2021	-
NCT01847716	Pulmonary arterial hypertension	Transforming Growth Factor Beta Signalling in the Development of Muscle Weakness in Pulmonary Arterial Hypertension	October 2013	-
NCT01893710	Peritoneal dialysis	International (Pediatric) Peritoneal Biobank	February 2011	-

## Data Availability

The datasets used and analyzed during the current study are available from the corresponding author on reasonable request.

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
