# Peer review of "The Role of TGF-β Signaling in Saphenous Vein Graft Failure after Peripheral Arterial Disease Bypass Surgery"

_ijms, 2023, doi:10.3390/ijms241210381_

Round 1
Reviewer 1 Report
This is a very interesting and well illustrated review. Only minor points can be improved. In particular:
Introduction: a bref introduction on TGFB signalling deserves to be added. Since this is a review article, the complexity of this signalling deserves to be highlighted. In fact, TGFB can favor cell motility, survival and proliferation (see PMID: 26708185 ). Moreover, TGFB induces collagen expression favoring fibrosis (see PMID: 32006713), a key process involved in peripheral arterial disease.
Figure 5: Authors should consider to split the images shown in this figure in two figures since the IHC images are too small and morphology and staining cannot be appreciated
An accurate revision of typing errors is recommended
References must follow the journal style
Author Response
Dear IJMS Editor and Reviewer,
On behalf of my co-authors, I am pleased to submit the revised manuscript entitled “The Role of TGF-β signaling in Saphenous Vein Graft Failure after Peripheral Arterial Disease Bypass Surgery” for consideration. We hope that you will find our manuscript of interest to yourself. Following your constructive suggestions, we had revised this manuscript. Thank you for considering this paper and please let us know if anything further is required.
Sincerely,
Chuanqi Cai.
Point 1:Introduction: a bref introduction on TGFB signalling deserves to be added. Since this is a review article, the complexity of this signalling deserves to be highlighted. In fact, TGFB can favor cell motility, survival and proliferation (see PMID: 26708185 ). Moreover, TGFB induces collagen expression favoring fibrosis (see PMID: 32006713), a key process involved in peripheral arterial disease.
Response 1: Thank you very much for your comment. We revised the manuscript by adding the contents and the reference. Thanks.
Point 2: Figure 5: Authors should consider to split the images shown in this figure in two figures since the IHC images are too small and morphology and staining cannot be appreciated
Response 2: We agree with your comment. We splited the Fig5 into two figures and revised the manuscript. Thanks.
Point 3: An accurate revision of typing errors is recommended
Response 3: We revised the whole manuscript. Thanks.
Point 4: References must follow the journal style
Response 4: We agree with this comment. We updated the endnote style for this manuscript. Thanks.

Reviewer 2 Report
The authors studied the “IThe Role of TGF-β signaling in Saphenous Vein Graft Failure after Peripheral Arterial Disease Bypass Surgery”. This study includes the mix between literature review, omics and case report. Please find below comments regarding this article:
1. Authors focus in peripheral arterial disease in graft failure mostly, however the graft stenosis is also seen during CABG. Why did the authors did not focus in this group of patient and related studies?
2. Why did the authors focused in such of small group of patient to detect the difference between the PAD and Control group? This seems more as a pilot study !!
3. TGF beta signalling is widely described in many experimental studies in saphenous vein, therefore I would recommend to authors to make a specific literature review regarding studies in saphenous vein more rather than describing the well known findings of TGB beta signalling. So adding the tables with Human Tissue and Animal are suggested to fullfill this.
4. What are current pharmacological approaches in targeting the TGF beta signalling inhibition? Authors need to extend this approach rather than mentioning some specific clinical trials. For example what about the role of statins? Authors need to focus to role of TGF signalling in atherosclerosis, PAD, grafts more.
5. What about the TGF-β signaling blockade drugs which are currently in development as clinical leads ?
6. The case report presented by the authors for my opinion is not very structured, however I would suggest the authors to presented in separately or either change the methodological approach of the study, from review to original article where there are some additional group of patients presented in similar manner. Specially to focus in TGF beta signalling in Control versus PAD patients.
7. There are studies involved in varicous vein versus normal, in which they study TGF beta signalling (significantly increased TGF-beta RII mRNA level was found in varicose veins when compared with normal veins).? How do authors comment this and the new addings and novelity of the study need to be emphasized more.
8. It has been reported that TGF signalling immunohistochemistry showed the presence of the transforming growth factor-β as in your case report? What about the Smad7 overexpressions in the extracellular matrix which may provide primary evidence for early or late graft failure.
9. The expression TGF-β1 in varicose veins was shown to be upregulated in the media tunica and intima tunica of varicose veins? What do the authors add with this case report?
10. How do authors add novelity beyond this study: Effect of TGF-beta1 on MMP/TIMP and TGF-beta1 receptors in great saphenous veins and its significance on chronic venous insufficiency?
11. As mentioned above there are different study approaches which needs to be studied and added for example: "Inhibition of transforming growth factor-β restores endothelial thromboresistance in vein grafts"
12. Authors skip describing the studies which replicate this case report "Persistently increased expression of the transforming growth factor-β1 gene in human vascular restenosis: Analysis of 62 patients with one or more episode of restenosis" or "Intimal hyperplasia and expression of transforming growth factor-beta1 in saphenous veins and internal mammary arteries before coronary artery surgery" So what is novel about this case report or added more to this study! If so please make a very careful literature review corresponding to the bioinformatics and recent finding interpretation accordingly.
Author Response
Dear IJMS Editor and Reviewer,
On behalf of my co-authors, I am pleased to submit the revised manuscript entitled “The Role of TGF-β signaling in Saphenous Vein Graft Failure after Peripheral Arterial Disease Bypass Surgery” for consideration. We hope that you will find our manuscript of interest to yourself. Following your constructive suggestions, we had revised this manuscript. Thank you for considering this paper and please let us know if anything further is required.
Sincerely,
Chuanqi Cai.
Point 1. Authors focus in peripheral arterial disease in graft failure mostly, however the graft stenosis is also seen during CABG. Why did the authors did not focus in this group of patient and related studies?
Response 1: We agree with your great comment. CABG failure is very important. Because all the coauthors in our team are from the vascular surgery department, we just focused on the PAD disease in this manuscript. However, we are co-working with our cardiac surgeons to study the graft failure in CABG and compare the differences between PAD graft failure and CABG failure. We added the CABG information in the Prospects section. Thanks.
Point 2. Why did the authors focused in such of small group of patient to detect the difference between the PAD and Control group? This seems more as a pilot study !!
Response 2: Yes, we agree with your comment. It’s a pity that the PAD data from the GEO database is not as fruitful as the CABG failure. So we tried to re-analyze the limited PAD database. And we completed around 25 PAD bypass surgeries and the following-up is undergoing. We will try to reveal the mechanism of PAD graft failure from human omics and IHC studies, and from the murine hide limb ischemia and vascular transplantation study. Thanks.
Point 3. TGF beta signalling is widely described in many experimental studies in saphenous vein, therefore I would recommend to authors to make a specific literature review regarding studies in saphenous vein more rather than describing the well known findings of TGB beta signalling. So adding the tables with Human Tissue and Animal are suggested to fullfill this.
Response 3: We agree with your comment. We added a table about the TGFB signaling in the saphenous vein study. Thanks.
Point 4. What are current pharmacological approaches in targeting the TGF beta signalling inhibition? Authors need to extend this approach rather than mentioning some specific clinical trials. For example what about the role of statins? Authors need to focus to role of TGF signalling in atherosclerosis, PAD, grafts more.
Response 4: We agree with your comment. However, there is no TGFB inhibiter used for the PAD bypass failure. We revised the table. Thanks.
Point 5. What about the TGF-β signaling blockade drugs which are currently in development as clinical leads ?
Response 5: We agree with your comment. We added this into the prospects section. Thanks.
Point 6. The case report presented by the authors for my opinion is not very structured, however I would suggest the authors to presented in separately or either change the methodological approach of the study, from review to original article where there are some additional group of patients presented in similar manner. Specially to focus in TGF beta signalling in Control versus PAD patients.
Response 6: We totally agree with your comment. The case report was a representative case to confirm the TGFB expression in our hypothesis. We will complete the original article when the three-years following-up are finished. Thank you very much for your suggestions.
Point 7. There are studies involved in varicous vein versus normal, in which they study TGF beta signalling (significantly increased TGF-beta RII mRNA level was found in varicose veins when compared with normal veins).? How do authors comment this and the new addings and novelity of the study need to be emphasized more.
Response 7: We agree with your meaningful comment. Yes, increased TGFB signaling was demonstrated in the varicose veins. Our comment is that the “healthier” vein graft may result in a better patency. Reduce the TGFB signaling in the vein graft may benefit the patency. And we revised the manuscript, thanks.
Point 8. It has been reported that TGF signalling immunohistochemistry showed the presence of the transforming growth factor-β as in your case report? What about the Smad7 overexpressions in the extracellular matrix which may provide primary evidence for early or late graft failure.
Response 8: We agree with your comment. In our PAD study, we found the patent graft vein showed lower TGF IHC staining (the project is going on). For Smad7, we agree with your comment, it’s important to complete the Smad7 staining. And we will complete the staining when all the tissue sections are ready, because the initial preoperative human tissues are precious. Thank you very much for your suggestions.
Point 9. The expression TGF-β1 in varicose veins was shown to be upregulated in the media tunica and intima tunica of varicose veins? What do the authors add with this case report?
Response 9: We agree with your comment. This implied that the TGF-β1 signaling my linked to several cellular types. Further experiments need to be designed to reveal the complex cellular biology. We revised this in the manuscript. Thanks.
Point 10. How do authors add novelity beyond this study: Effect of TGF-beta1 on MMP/TIMP and TGF-beta1 receptors in great saphenous veins and its significance on chronic venous insufficiency?
Response 10: This is a very good suggestion. We added this into the prospects section. Thanks.
Point 11. As mentioned above there are different study approaches which needs to be studied and added for example: "Inhibition of transforming growth factor-β restores endothelial thromboresistance in vein grafts"
Response 11: We agree with your comment. We added this reference. Thanks.
Point 12. Authors skip describing the studies which replicate this case report "Persistently increased expression of the transforming growth factor-β1 gene in human vascular restenosis: Analysis of 62 patients with one or more episode of restenosis" or "Intimal hyperplasia and expression of transforming growth factor-beta1 in saphenous veins and internal mammary arteries before coronary artery surgery" So what is novel about this case report or added more to this study! If so please make a very careful literature review corresponding to the bioinformatics and recent finding interpretation accordingly.
Response 12: We agree with your comment. We are sorry for this. We added these two excellent references. The case is from one of our PAD graft failure project, and we used this case to confirm the TGFB expression corresponding to the bioinformatics data. Thank you very much.
